# Three-Dimensional Transesophageal Echocardiography in the Diagnosis and Treatment of Mitral Prosthetic Valve Endocarditis—A Narrative Review

**DOI:** 10.3390/medicina58010023

**Published:** 2021-12-23

**Authors:** Vedran Carević, Zorica Mladenović, Ružica Perković-Avelini, Tina Bečić, Mislav Radić, Damir Fabijanić

**Affiliations:** 1Department of Cardiology, Clinical Hospital Centre Split, Spinčićeva 1, 21000 Split, Croatia; vcarevic@yahoo.com (V.C.); ruzica.perkovic.avelini@gmail.com (R.P.-A.); tina.becic@gmail.com (T.B.); 2Department of Internal Medicine, University of Split School of Medicine, Šoltanska 2, 21000 Split, Croatia; mislav.radic@gmail.com; 3Clinic of Cardiology, Military Medical Academy, Crnotravska 17, 11040 Belgrade, Serbia; zoz3377@gmail.com; 4Division of Rheumatology, Clinical Hospital Centre Split, Spinčićeva 1, 21000 Split, Croatia

**Keywords:** infective endocarditis, mitral valve prosthesis, prosthetic valve endocarditis, three-dimensional transesophageal echocardiography

## Abstract

Despite advances in diagnosis, imaging methods, and medical and surgical interventions, prosthetic valve endocarditis (PVE) remains an extremely serious and potentially fatal complication of heart valve surgery. Characteristic changes of PVE are more difficult to detect by transthoracic echocardiography (TTE) than those involving the native valve. We reviewed advances in transesophageal echocardiography (TEE) in the diagnosis of PVE. Three-dimensional (3D) TEE is becoming an increasingly available imaging method combined with two-dimensional TEE. It contributes to faster and more accurate diagnosis of PVE, assessment of PVE-related complications, monitoring effectiveness of antibiotic treatment, and determining optimal time for surgery, sometimes even before or without previous TTE. In this article, we present advances in the treatment of patients with mitral PVE due to 3D TEE application.

## 1. Introduction

Prosthetic valve endocarditis (PVE) remains an extremely serious and potentially fatal complication of heart valve surgery, associated with a mortality risk of about 20–30% [1,2]. Its incidence is 0.3–1.2% cases per patient-year and affects 1–6% of patients with a prosthetic valve (PV) [3,4]. Mitral PVE is less common than aortic, with an incidence in mitral biological valves of 0.49% per patient-year (vs. 0.91% for aortic valves) and in mitral mechanical valves of 0.18% per patient-year (vs. 0.27% for aortic valves) [3,4].

According to the time of infection occurrence since valve replacement surgery and specific microbiology, PVE can be classified into two types: early (usually a manifestation of healthcare-acquired infection; most frequently caused by *Staphylococcus aureus*, less commonly by coagulase-negative *Staphylococcus species*) and late (usually acquired in the community and caused by *Streptococcus viridans* or the HACEK group of bacteria) [5]. The time interval between the two mentioned PVE types is considered to be 1 year [5].

More than two-thirds of PVE occurs during the first year after valve replacement [4,5]. Mechanical prostheses are at a higher risk of early infection than bioprostheses mainly due to insufficient endothelization [4,5]. In contrast, bioprostheses have a greater predisposition to late PVE primarily due to leaflet degeneration over time [4,5]. Nevertheless, the cumulative risk of PVE appears to be similar in both types of PV [4].

Treatment of PVE is directed at eradication of the infective agent with different antibiotic protocols, effective in about 50% of patients [1,5]. Cardiac surgery in patients with PVE is usually recommended after a standard course of antibiotic therapy in case of signs of uncontrolled infection, hemodynamic instability, and a high risk of embolic complications [1,5].

According to current guidelines, echocardiography is considered the basic diagnostic method of PVE [5,6]. However, PVE is more difficult to detect by transthoracic echocardiography (TTE) than infective endocarditis (IE) of native valves; hence, transesophageal echocardiography (TEE) is a recommended diagnostic tool in PVE [1,5,6,7,8]. Three-dimensional (3D) TEE is becoming an increasingly available echocardiographic modality in routine clinical practice. With different types of datasets and the ability to manipulate and crop images in postprocessing, 3D TEE enables an analysis of deep and sheltered anatomical structures by providing additional valuable information in PVE assessment. Moreover, 3D TEE visualization is not limited to conventional 2D planar views. The possibility for further rotation and angulation allows 3D TEE assessment of any heart structure from numerous planes and angles. 3D TEE en face view is particularly useful in diagnosing endocarditis of mitral valve prosthesis. This echocardiographic presentation of mitral PV allows detection of discrete changes, such as small PVE vegetation or discrete valvular dehiscence [8,9,10,11,12,13].

In this review, we present advances in the diagnosis of mitral PVE patients due to 3D TEE application.

## 2. D TEE vs. 3D TEE in Mitral PVE Treatment

Two-dimensional TTE sensitivity and specificity for diagnosis of PVE are low due to limitations caused by acoustic shading, body habitus, and possible concomitant pulmonary pathology [1,5,6,7,8]. Two-dimensional TEE has been shown to be superior to 2D TTE in diagnosing PVE, especially in a case of posteriorly placed mitral prosthesis [1,7]. The position of the TEE probe near the left atrium allows enhanced visualization of the left atrial side of the mitral prosthesis and more precise visualization of vegetation, abscess, paravalvular regurgitation, and dehiscence of prosthetic mitral valve [1,7]. According to the literature, in the assessment of mitral PV pathology TEE has an overall sensitivity of 96% [7]. Hence, TEE is indicated in all patients with suspected mitral PVE, sometimes even before or without previous TTE [1,7].

According to recent reports, 3D TEE provides clear, rapid, and detailed anatomical visualization of mechanical and bioprosthetic mitral valves in real time ensuring better assessment of mitral PVE-related complications and significant improvement in monitoring and preoperative surgical planning [9,10,11,12,13,14]. The surgeon’s view, which shows the mitral prosthesis with the aortic root at the 12 o’clock position, is identical to what the surgeon sees and is recommended as a standard orientation (Figure 1) [9,13]. Therefore, 3D TEE precise anatomical imaging of the mitral valve is better understood by cardiac surgeons [8]. An additional advantage of 3D TEE is better visualization of the mitral prosthesis from the left ventricle, although shading caused by a mechanical prosthesis can sometimes make its clear visualization difficult or impossible [8]. Finally, 3D TEE echocardiographic modality is capable of improving communication between members of the IE team.

## 3. Echocardiographic Presentation of the Mitral PVE

The ‘Duke criteria’, introduced in 1994 and modified in 2000, formalized the role of echocardiography in the diagnostic strategy of IE [15,16]. Echocardiographic findings suggestive of IE were accepted as one of the major diagnostic criteria. Three echocardiographic features represent the main criteria for PVE:oscillating intracardiac mass in accordance with vegetation,abscess formation,presence of paravalvular regurgitation/leakage or dehiscence of prosthetic valve.

Echocardiographers should keep in mind that the pathophysiology and echocardiographic presentation of PVE have a certain different specificity in relation to the type of prosthesis and the time elapsed since valve replacement. In mechanical PVE, infection usually begins early after replacement from the prosthetic ring suture line or from a nonbacterial thrombotic lesion near the recirculation flow area or prosthetic ring, leading to complications associated with a worse prognosis than late PVE requiring valve replacement [2,3,5]. Because metal and pyrolytic carbon are not suitable for adhesion of microorganisms, late PVE on a mechanical prosthesis is most commonly formed during bacteriemia by adhesion of bacteria to thrombotic material formed on the surface of the pannus [2,3,17]. Infection of the bioprosthesis usually occurs in later years and has a low tendency to attack annular and perianular tissues, but is mostly confined to the leaflet body, primarily in areas of tissue degeneration that predispose colonization by microorganisms [3,17]. This infection mainly responds to antibiotic therapy, although degeneration and erosion of leaflets can cause acute and late valve failure. In case an infection of the bioprosthesis affects the annular and perivalvular tissues, abscesses may appear and then the clinical course of the disease is similar to that of a mechanical prosthesis [2,3].

### 3.1. PVE Vegetation

With TTE, it is more difficult to detect vegetation on PV than that involving native valves, because the highly echogenic suture ring and PV support structures can prevent detection of vegetation within the valve apparatus or its shadow [6,7]. Therefore, TEE should always be used if a diagnosis of PVE is suspected. In large series of PVE, TEE showed 86–94% sensitivity and 88–100% specificity for vegetation diagnosis; at the same time, the sensitivity of TTE was 36–69% [6,7]. Vegetative growth usually occurs as an abnormal echogenic mass with independent free movement attached to the atrial side of the mitral PV (Figure 2) [6,7].

However, vegetative growth can occur as thickening and irregularity of the normally smooth contour of the suture ring (Figure 2) or in the case of biological PVE as vegetation attached to the leaflets (Figure 3), sometimes with their consequent perforation or creation of a pseudoaneurysm [6,7].

According to the European Society of Cardiology guidelines, there are three main criteria for indicating emergent surgery to prevent embolism on the left PVE: PV with vegetation >10 mm after a previous embolic episode, very large isolated vegetation (>30 mm), and isolated vegetation with main diameter >15 mm and no other indication for surgery [5].

Because the risk of embolism in patients with IE is related to the size, morphology, and mobility of vegetation, proper assessment of vegetation and monitoring of their size during antibiotic therapy have important prognostic implications [10,11]. Therefore, the accuracy and reproducibility of vegetation measurements is particularly important due to the greater effectiveness of early surgical interventions in patients with large mobile vegetation [10,11]. In this context, it is important that 2D TEE could underestimate vegetation compared to 3D TEE primarily due to limitations in selecting the maximum true diameter of irregular masses [10,11]. This shortcoming is solved by 3D TEE which allows infinite planes and volumetric reconstruction of masses (Figure 2). Because the main diameter of vegetation is important in the assessment of medical treatment response, as well as for setting a surgical indication in patients without other surgical indications, differences between the two TEE techniques in accuracy of vegetation measurements may have key therapeutic implications. In addition, 3D TEE allows better morphological classification of vegetation (e.g., filiform, sedentary, and clustered), and recognition of those associated with fragility and embolization (it seems that sedentary vegetation has the least embolic potential) [10,11]. Therefore, the main advancement of 3D TEE over 2D TEE is better assessment of vegetation characteristics—size, location, number, morphology, consistency, mobility, and relationship to valve structure—resulting in better prediction of potential embolic risk [10,11].

It is obvious that by providing better images in comparison to 2D TEE, 3D TEE allows detailed anatomical visualization of PV and better characterization of vegetation allowing their differentiation from non-PVE changes such as thrombi, pannus, prosthetic sutures, mitral subvalvular tissue remnants (Figure 1d,e), and bioprosthesis leaflet degeneration [18]. Since these findings have a very similar echocardiographic appearance, sometimes it is not possible to distinguish them from each other only by echocardiographic parameters.

### 3.2. Perivalvular Complications

Perivalvular complications are particularly common in early PVE because the annular ring is usually the primary site of infection [1,4]. The spread of infection to perivalvular tissues is a sign of a poor prognosis in the evolution of the disease and represents the main criteria for the diagnosis of PVE and an indication for urgent surgical intervention [1,5].

Perivalvular abscess is more often associated with PVE of the aortic valve than with mitral PVE and is thought to be present when a specific area of reduced echo density is found on the echocardiogram or when echolucent cavities are found within the valvular ring or myocardial structures surrounding the valvular infection [7]. In case of mitral PV, the spread of infection to perivalvular tissue is more often manifested as new or increased paravalvular regurgitation (Figure 4) or valve dehiscence (Figure 5) [4]. New regurgitation in patients with IE of bioprosthesis detected several years after valve replacement is primarily caused by perforation or rupture of the leaflets with consequent varying degrees of insufficiency and only rarely by paravalvular mechanisms [4,17].

Although 2D TEE is a useful diagnostic tool in detection of paravalvular complications (sensitivity 90% and specificity 90%), 3D TEE provides anatomical imaging that can better identify location, number, shape, and size of damage [1,13]. Recent studies with 3D TEE found that mitral PV dehiscence is most frequently located in the posterior and lateral part of mitral annular region [1,13]. In relation to the extent of the affected PV ring, the size of the defect can be classified into three categories: mild (<10%), moderate (10–29%) and severe (>30%) [13]. If the PV dehiscence exceeds 40% of the valve ring, rocking the prosthesis can cause severe regurgitation and precipitate congestive heart failure. In addition, when assessing significance of a defect, its shape and surface should be taken into account [13]. Namely, a circular defect with a fairly large surface area occupying less than 10% of the ring circumference may be hemodynamically more significant than a slit-shaped defect with a small surface area occupying more than 20% of the circumference (Figure 4).

## 4. Advantages and Limitations of 3D TEE in PVE Diagnostics

Diagnosis of PVE is challenging and includes identification of the infective pathogen and detection of cardiac lesions and noncardiac localization of the disease. Evidence of intracardiac material involvement is a major diagnostic criterion, with TEE representing the first-line imaging method [5]. Higher quality of 3D compared to 2D images allows a more realistic anatomical view and a better assessment of changes characteristic of PVE, thus, providing additional information important for fast and accurate diagnosis and timely therapeutic decisions (Table 1).

Three-dimensional TEE has several limitations that include poor visualization of anterior cardiac structures, such as aortic and tricuspid valves, while mitral valve as a posterior structure due to reduced distance from TEE probe can be visualized significantly better; suboptimal images due to poor ECG triggering in patients with arrhythmias; reduced spatial and temporal resolution with narrow angled acquisitions; and tissue dropout [19]. These limitations can be addressed with newer echocardiography technology and training. Other limitations are common to 2D and 3D TEE and include nondetection of peripheral complications (metastatic infection and infectious embolism) and potential procedural complications [5,20]. In situations where the diagnosis of PVE is uncertain with echocardiography, anatomic imaging such as multislice computed tomography (CT) or hybrid modalities, which combine both anatomic and metabolic information such as positron emission tomography (PET)/CT and single photon emission tomography (SPECT), has been shown to be particularly useful [20]. These techniques have gradually been included in the diagnostic algorithm in assessment of patients with suspected PVE despite the use of these imaging techniques has not yet been standardized and varies widely in clinical practice [5,20].

## 5. Conclusions

Two-dimensional TEE detects signs of mitral PVE with excellent sensitivity and specificity. However, the higher quality of 3D compared to 2D images allows a more realistic anatomical view and a delineation of valvular vegetation and abscesses, perforations and paravalvular leaks, dehiscence, and localization of infection with regard to valve anatomy and its relationship with surrounding cardiac structures. Those providing additional information important for fast and accurate diagnosis and timely therapeutic decisions. Finally, the ability of 3D TEE to accurately display mitral PV from a left atrial perspective identical to that seen by the surgeon during surgery (surgeon’s view) allows better communication within the IE team.

Conclusively, 3D TEE imaging improves visualization in the assessment of mitral PVE in terms of prognosis and selection of the best therapeutic options; as is the least costly and most affordable method.

## Figures and Tables

**Figure 1 medicina-58-00023-f001:**
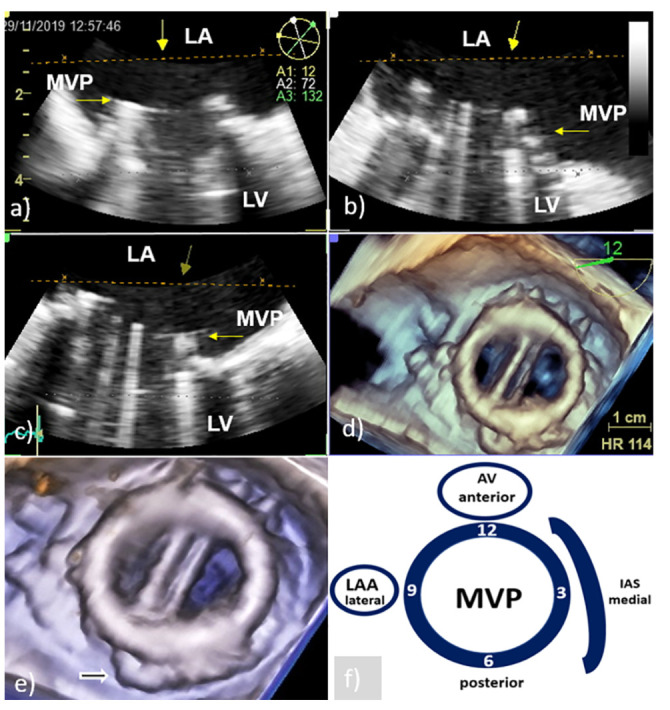
Transesophageal echocardiography of MVP: Presentation of the MVP in different two-dimensional planes needs careful mental reconstruction to translate the echo description into the anatomy (**a**–**c**); 3D TEE reconstruction allows realistic anatomic image of the MVP similar to the one the surgeon sees from the LA (surgeon’s view) (**d**); cropped and enlarged 3D TEE image allows better presentation of the MVP and surrounding structures (white arrow—residual tissue of posterior leaflet of native mitral valve) (**e**); schematic presentation of surgeon’s view (clockwise orientation) with AV anteriorly at 12 o’clock position, IAS at the middle 3 o’clock position, and LAA at the lateral 9 o’clock position (**f**). AV—aortic valve; IAS—interatrial septum; LA—left atrium; LAA- left atrium appendage; LV—left ventricle; MVP—mitral valve prosthesis; and 3D TEE—three-dimensional echocardiography.

**Figure 2 medicina-58-00023-f002:**
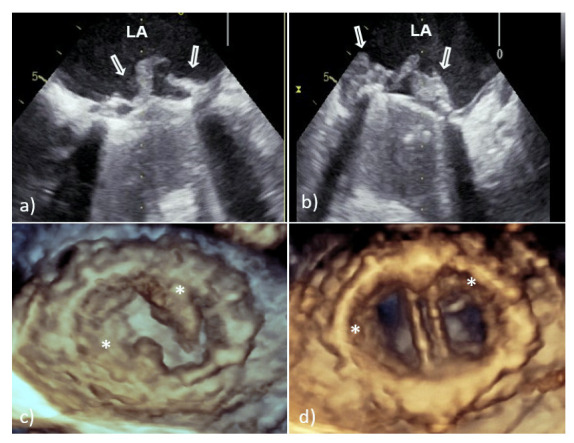
Transesophageal echocardiography of mechanical MVP endocarditis caused with *Staphylococcus aureus*: 2D TEE showed an extensive vegetation (white arrows) attached to the prosthetic ring and leaflets (**a**,**b**); 3D TEE showed voluminous vegetation (asterisks) affecting a whole circumference of sewing ring and MVP leaflets (**c**—in systole, **d**—in diastole). 2D TEE—two-dimensional transesophageal echocardiography; 3D TEE—three-dimensional transesophageal echocardiography; LA—left atrium; PVE—prosthetic valve endocarditis; and MVP—mitral valve prosthesis.

**Figure 3 medicina-58-00023-f003:**
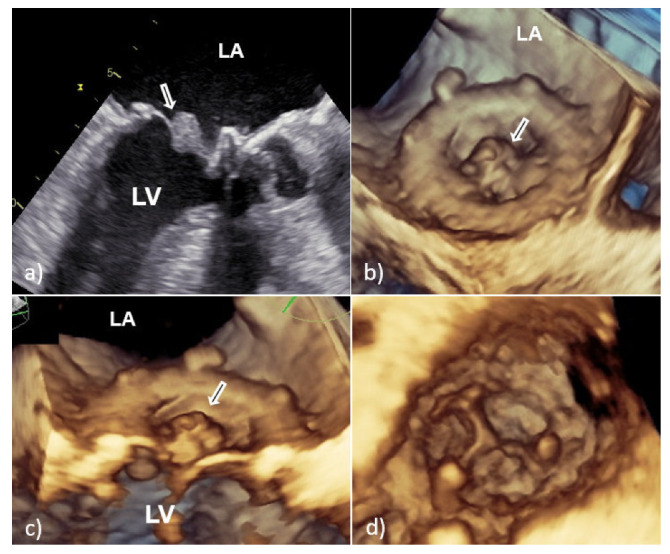
Transesophageal echocardiography of mitral bioprosthesis endocarditis caused by *Enterococcus faecalis*: 2D TEE (**a**) and 3D TEE view (**b**,**c**) from the atrial side showed a sedentary type of vegetation (white arrow) clinging to the leaflet with a wide base; view from the left ventricle showed good leaflet appearance and coaptation with no possible complications of PVE (**d**). 2D TEE—two-dimensional transesophageal echocardiography; 3D TEE—three-dimensional transesophageal echocardiography; PVE—prosthetic valve endocarditis; LA—left atrium; and LV—left ventricle.

**Figure 4 medicina-58-00023-f004:**
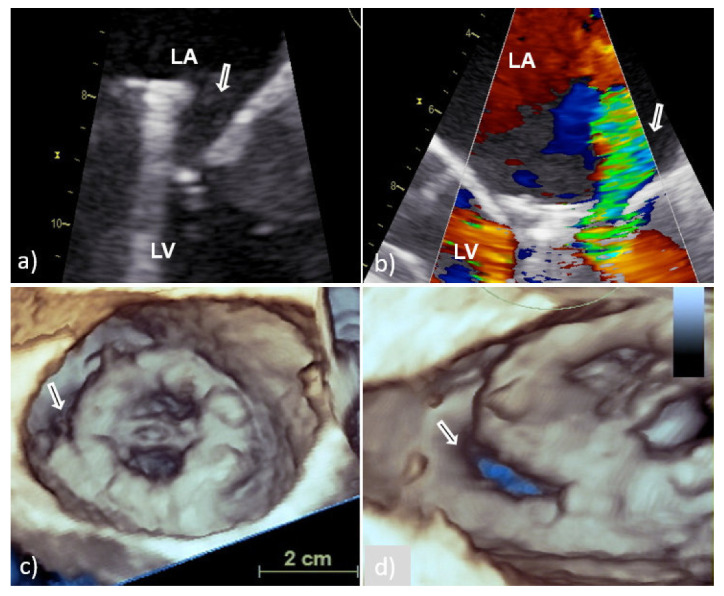
Transesophageal echocardiography of MVP endocarditis caused by *Streptococcus bovis*. Two-dimensional TEE shoved localized dehiscence of the MVP with a paravalvular leakage (white arrow) (**a**,**b**); surgeon’s view of MVP with suspected dehiscence (white arrow) in the lateral (8–9 o’clock) position (**c**); enlarged and rotated 3D TEE image from different angle confirms a defect (white arrow) with fairly large surface area that occupies approximately 15% of the MVP circumference and causes hemodynamically significant regurgitation (**d**). 2D TEE—two-dimensional transesophageal echocardiography; 3D TEE—three-dimensional transesophageal echocardiography; MVP—mitral valve prosthesis; LA—left atrium; and LV—left ventricle.

**Figure 5 medicina-58-00023-f005:**
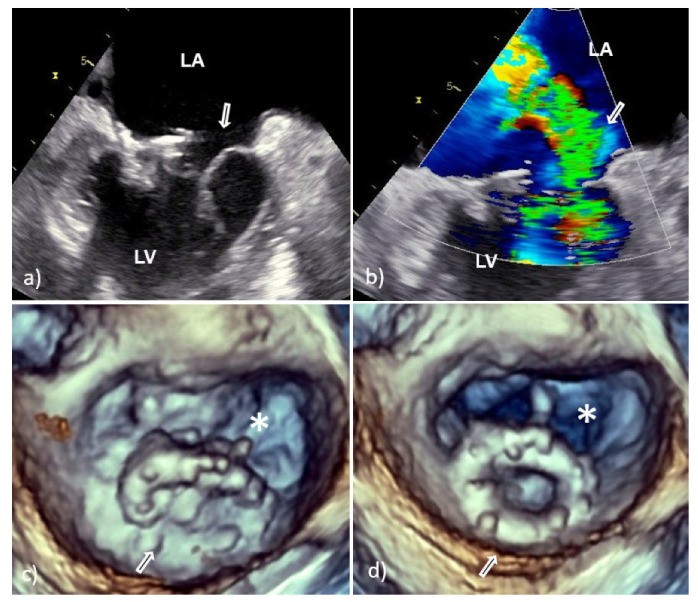
Transesophageal echocardiography of the mitral valve 15 days after surgical correction due to mitral valve prolapse. Two-dimensional TEE revealed dehiscence of the C-E ring with consequent severe regurgitation (white arrow, **a**,**b**); 3D TEE showed almost complete dehiscence of the C-E ring which was attached to the native ring only on the smaller posterolateral segment (white arrow); the asterisk indicates the surface of the native mitral orifice (**c**—in systole, **d**—in diastole). Clinical and laboratory parameters along with *Staphylococcus epidermidis* revealed in blood culture suggest that dehiscence was caused by infective endocarditis. 2D TEE—two-dimensional transesophageal echocardiography; 3D TEE—three-dimensional transesophageal echocardiography; C-E—Carpentier-Edwards ring; LA—left atrium; and LV—left ventricle.

**Table 1 medicina-58-00023-t001:** Comparison between 2DTEE and 3DTEE in diagnosing of different prosthetic valve endocarditis pathologies (modified according [8]).

	3D TEE vs. 2D TEE
PV vegetation	
Identification	superior
number	superior
size	superior
location	superior
attachment	superior
PV abscess	
identification	equal
size	superior
site	superior
extension	superior
communication	superior
PV regurgitation	
transvalvular	equal
paravalvular	superior
mechanism	superior
severity	equal, both satisfying
jet direction	equal, both satisfying
PV dehiscence	
identification	superior
size	superior
site	superior
shape	superior
area	superior
relation to leak	superior

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
