# Peer review of "Three-Dimensional Transesophageal Echocardiography in the Diagnosis and Treatment of Mitral Prosthetic Valve Endocarditis—A Narrative Review"

_medicina, 2021, doi:10.3390/medicina58010023_

Round 1

Reviewer 1 Report

Line 79-84, The authors should clarify whether these conclusions are supported by data or they are based on experience of the authors (and logic to a certain point). If there is data to support them, please reference it.

If all parts of figure 1 are obtained from the same patients, I suggest to authors to make all these figures bigger and perhaps mark certain referencing points on figures a-c, to make them more readable (and in fact to further address the challenging nature of 2D TEE interpretation, as the authors mentioned). 

Line 99. Adjust styling of numbering

Line 105.  Did the authors mean differences in specificity in relation to…?

Since Table 1 is modified according to ref No.8, I wonder if the authors obtained a permission from the authors of original table

Overall, the authors did not mention any shortcoming of the 3D TEE. If there are any, please clarify them in the text. In addition, the authors should present some future perspectives, as in this way I see no much novelty in the present review.

Author Response

We thanking you for taking the time to assess our manuscript and thank you for the constructive comments on it.

We addressed all the concerns that raised:

Point 1: Line 79-84, The authors should clarify whether these conclusions are supported by data or they are based on experience of the authors (and logic to a certain point). If there is data to support them, please reference it.

Response 1: We accept your comment. These conclusions are based on authors' experience and from evidence of literature. So we reffered that to emphasize this point.

Point 2: If all parts of figure 1 are obtained from the same patients, I suggest to authors to make all these figures bigger and perhaps mark certain referencing points on figures a-c, to make them more readable (and in fact to further address the challenging nature of 2D TEE interpretation, as the authors mentioned

Response 2: Thank you for this suggestion. We have enlarged and additionally marked the figures a-c of this particular patient and we hope that it is now easier to recognize the structures in the images since our main idea was to show a successful 3D reconstruction from these blurred images.

Point 3: Line 99. Adjust styling of numbering.

Response 3: Thanks for noticing this. We’ve corrected the typo.

Point 4: Line 105. Did the authors mean differences in specificity in relation to…?

Response 4: We have made the change. The new sentence reads as follows: Echocardiographers should keep in mind that the pathophysiology and echocardiographic presentation of PVE have a certain different specificity in relation to the type of prosthesis and the time elapsed since valve replacement.

Point 5: Since Table 1 is modified according to ref No.8, I wonder if the authors obtained a permission from the authors of original table

Response 5: We understand the reviewer's concern regarding the copyright. We would like to point out that we have not reproduced the original work. We appreciate any specific example that could help us improve the clarity of the manuscript. We took the structure of the table, which we referred to, and made changes in the context of this review article.

Point 6: Overall, the authors did not mention any shortcoming of the 3D TEE. If there are any, please clarify them in the text. In addition, the authors should present some future perspectives, as in this way I see no much novelty in the present review.

Response 6: We appreciate the reviewer’s insightful suggestion and agree that it would be useful to demonstrate that 3D TEE has several limitations on PVE diagnosing. We are aware of all its shortcomings, but also on other imaging modalities and theirs evidence-based validations. We are trying to target both readers that are interested in the comprehensive study and readers that are only interested in subsections. Therefore we have revised the text to address your concerns, which including new section “4. Advantages and limitations of 3D TEE in PVE diagnostics”. Please see page 8 of the revised manuscript, lines 270-278; and page 9 lines 293-310, lines 313-319, and lines 323-325.

We would like to thank you once again for reviewing our manuscript and the opportunity to improve it with your comments. We sincerely hope that the revised manuscript can be accepted for publication.

Reviewer 2 Report

Dear authors, in my opinion, every TOE includes 3D imaging. There are already new and specific 3D technologies inliuding cardiac true vue or the echo navigator. Therefore, I don't think that 3D TOE imaging is that special in the assessment of endocarditis. What about nuclear imaging additional to TOE 2D/3D?

Author Response

We thanking you for taking the time to assess our manuscript, and for your comments on it.

Point: Dear authors, in my opinion, every TOE includes 3D imaging. There are already new and specific 3D technologies including cardiac true vue or the echo navigator. Therefore, I don't think that 3D TOE imaging is that special in the assessment of endocarditis. What about nuclear imaging additional to TOE 2D/3D?

Response: You have raised an important point here. We agree that new advanced medical imaging software and artificial intelligence could provide better insight and would be helpful as new echocardiographic tools. However, those methods, due to the costs and equipment involved, even in case of 3D transesophageal echocardiography, are unfortunately not affordable for most practitioners from developing countries such ours. Also, we agree that nuclear medicine procedures gave us an important contribution to the diagnostic assessment of endovascular infections. The context of our review article was not intended to compare echocardiography with other imaging methods, but we are aware of all its shortcomings. Therefore we have revised the text to address your concerns and hope that it is now clearer. We added section “4. Advantages and limitations of 3D TEE in PVE diagnostics Please see page 8 of the revised manuscript, lines 270-278; and page 9 lines 293-310, lines 313-319, and lines 323-325.

We would like to thank again for reviewing our manuscript and appreciate your comments. We sincerely hope that this novelty improve manuscript and that can be accepted for publication.

Reviewer 3 Report

I want to thank the handling editor for providing me the opportunity to review this informative paper. I would also like to congratulate the authors for their elegant manuscript and comprehensive review.

This is a narrative review on the advances of transoesophageal echocardiography in the diagnosis of prosthetic valve endocarditis. The authors conclude that three-dimensional transoesophageal echocardiography can contribute to faster and more accurate diagnosis of mitral prosthetic valve endocarditis, offer improved monitoring of the response to medical treatment, and guide decisions regarding the optimal time for surgery.

The authors discuss the topic in an unbiased manner, covering its most significant aspects. Furthermore, the conclusions drawn are a logical sequence of the content and help identify areas for future research. Finally, the references are appropriate and related to the recent literature.

Author Response

We appreciate your positive feedback and thanking you for taking the time to assess our manuscript.

You wrote: I want to thank the handling editor for providing me the opportunity to review this informative paper. I would also like to congratulate the authors for their elegant manuscript and comprehensive review. This is a narrative review on the advances of transoesophageal echocardiography in the diagnosis of prosthetic valve endocarditis. The authors conclude that three-dimensional transoesophageal echocardiography can contribute to faster and more accurate diagnosis of mitral prosthetic valve endocarditis, offer improved monitoring of the response to medical treatment, and guide decisions regarding the optimal time for surgery. The authors discuss the topic in an unbiased manner, covering its most significant aspects. Furthermore, the conclusions drawn are a logical sequence of the content and help identify areas for future research. Finally, the references are appropriate and related to the recent literature.

Response: Due to suggestions of other reviewers about considerations of shortcomings of 3D transesophageal echocardioography we added section “4. Advantages and limitations of 3D TEE in PVE diagnostics”. Please see page 8 of the revised manuscript, lines 270-278; and page 9 lines 293-310, lines 313-319, and lines 323-325.

We would like to thank again for reviewing our manuscript and appreciate your comments. We sincerely hope that this novelty improve manuscript and that can be accepted for publication.

Round 2

Reviewer 1 Report

No further comments.

Reviewer 2 Report

Dear authors, coming from a highest level hospital, 3D TOE is our daily business. But I understand that this technique is  not that natural in other countries. Nevertheless, in my mind the manuscript has significantly improved due to your revisions.